# T- and B-Cells in the Inner Invasive Margin of Hepatocellular Carcinoma after Resection Associate with Favorable Prognosis

**DOI:** 10.3390/cancers14030604

**Published:** 2022-01-25

**Authors:** Andriy Trailin, Lenka Červenková, Filip Ambrozkiewicz, Esraa Ali, Phanindra Kasi, Richard Pálek, Petr Hošek, Vladislav Třeška, Ondrej Daum, Zbyněk Tonar, Václav Liška, Kari Hemminki

**Affiliations:** 1Laboratory of Translational Cancer Genomics, Biomedical Center, Faculty of Medicine in Pilsen, Charles University, Alej Svobody 1665/76, 32300 Pilsen, Czech Republic; filip.ambrozkiewicz@lfp.cuni.cz (F.A.); esraa.ali@lfp.cuni.cz (E.A.); kari.hemminki@lfp.cuni.cz (K.H.); 2Laboratory of Cancer Treatment and Tissue Regeneration, Biomedical Center, Faculty of Medicine in Pilsen, Charles University, Alej Svobody 1665/76, 32300 Pilsen, Czech Republic; lenka.cervenkova@lfp.cuni.cz (L.Č.); palekr@fnplzen.cz (R.P.); petr.hosek@lfp.cuni.cz (P.H.); liskav@fnplzen.cz (V.L.); 3Department of Pathology, Third Faculty of Medicine, Charles University, Ruská 87, 10000 Prague, Czech Republic; 4Department of Medical Chemistry and Biochemistry, Faculty of Medicine in Pilsen, Charles University, Karlovarská 48, 30166 Pilsen, Czech Republic; phanindra.kasi@lfp.cuni.cz; 5Department of Surgery and Biomedical Center, Faculty of Medicine in Pilsen, Charles University, Alej Svobody 80, 32300 Pilsen, Czech Republic; treska@fnplzen.cz; 6Sikl’s Institute of Pathology, Faculty of Medicine and Teaching Hospital in Plzen, Charles University, ul. Dr. E. Beneše 13, 30599 Pilsen, Czech Republic; daum@fnplzen.cz; 7Bioptická Laboratoř s.r.o., Mikulášské Nám. 4, 32600 Pilsen, Czech Republic; 8Department of Histology and Embryology, Faculty of Medicine in Pilsen, Charles University, Karlovarská 48, 30166 Pilsen, Czech Republic; zbynek.tonar@lfp.cuni.cz; 9Laboratory of Quantitative Histology, Biomedical Center, Faculty of Medicine in Pilsen, Charles University, Alej Svobody 1665/76, 32300 Pilsen, Czech Republic; 10Department of Cancer Epidemiology, German Cancer Research Center, Im Neuenheimer Feld 280, 69120 Heidelberg, Germany

**Keywords:** hepatocellular carcinoma, tumor-infiltrating lymphocytes, T-cells, B-cells, tumor invasive margin, stereology, heterogeneity, prognosis, time to recurrence, disease-free survival

## Abstract

**Simple Summary:**

Hepatocellular carcinoma (HCC) is one of the most common cancers in the world, which frequently recurs after curative resection. Several options to predict recurrence of HCC have been proposed, however, their prognostic ability is limited. This study aimed to test the hypothesis that distribution and numbers of T- and B-lymphocytes in different regions of the resected tumor may have different prognostic significance. Different subregions of HCC demonstrated uneven lymphocyte infiltration. CD20+ B-lymphocytes and CD8+ T-lymphocytes, or their combination in the inner tumor invasive margin and inner/outer margin ratios, convey the best prediction for time to recurrence and disease-free survival. The results offer a novel approach to the stratification of the risk of early tumor recurrence after curative liver resection.

**Abstract:**

In this retrospective study on 67 patients with hepatocellular carcinoma (HCC), after tumor resection, we evaluated the significance of CD3+ and CD8+ T-lymphocytes and CD20+ B-lymphocytes in tumor and non-tumor liver for time to recurrence (TTR), disease-free survival (DFS) and overall survival. After immunohistochemical staining, the density of nucleated lymphocyte profiles (Q_A_) was estimated stereologically in the tumor center (TC), inner margin (inn M), outer margin (out M), peritumor and non-tumor liver. In TC, intermediate and high Q_A_ of CD8+ cells predicted longer TTR, whereas CD3+ and CD20+ were predictive only at high Q_A_. DFS was predicted by high Q_A_ of CD3+, CD8+ and CD20+ cells in TC. The inn M harbored smaller Q_A_ of CD3+, CD8+ and CD20+ lymphocytes than out M. In contrast to out M, high T-cells’ Q_A_ and intermediate and high B-cell Q_A_ in inn M predicted longer TTR and DFS. High inn M/out M Q_A_ ratios of CD3+ and CD20+ cells were associated with longer TTR and DFS, whereas high inn M/out M Q_A_ ratio of CD8+ was predictive only for DFS. Patients with intermediate-high Q_A_ of combined CD8+ and CD20+ cells in inn M showed longer TTR and DFS, compared to CD8+-high or CD20+-high alone. Our findings highlight overall heterogeneity of the tumor invasive margin, the importance of inn M, and the predictive role of B-cells.

## 1. Introduction

Liver cancer, which in 75–90% of cases is represented by hepatocellular carcinoma (HCC), ranked as the sixth most common cancer in the world in 2020 and is the third leading cause of cancer death worldwide [1]. It is estimated that more than 1 million people will suffer from liver cancer each year by 2025 [2]. Liver resection and transplantation are the main therapeutic modalities for the treatment of HCC [3].

Several options to predict recurrence of HCC have been proposed, including TNM staging, microvascular invasion, tumor multiplicity, serum α-fetoprotein level, satellitosis [4,5] or neutrophil-lymphocyte ratio (reviewed in Najjar et al. [6]). However, their prognostic ability is limited and non-standardized [7,8]. Patients diagnosed with the same stage of disease often have markedly different outcomes, which can be related to varying involvement of endogenous anti-tumor mechanisms [9]. Recently, considerable data from large cohorts of various tumors have demonstrated that the assessment of number, type, location and functional orientation of tumor-infiltrating lymphocytes (TIL), together termed the tumor immune microenvironment (TIME), could improve the prediction of clinical outcome [10].

For colorectal cancer, the Immunoscore, based on the quantification of CD3+ and CD8+ T-cells in the tumor center (TC) and in the advancing tumor margin (margin), has turned out to be a better predictor of patient survival than histopathological methods, TNM staging, lymphovascular invasion, tumor differentiation or microsatellite instability status [11]. Immunoscore was further confirmed to be beneficial for the prognosis of disease-free survival (DFS) and overall survival (OS) in many different cancers. However, the predictive role of infiltration by adaptive immune cells in patients with HCC, who underwent resection, remains less clear. A vast majority of studies evaluated only intratumor lymphocytes (reviewed in Ding et al. [12]), whereas tumor margin attracted less attention. At the same time, guidelines for the assessment of TILs in solid tumors recommend separate reporting of cell densities in TC and the invasive margin, and designate margin as a 1 mm region centered on the border separating the malignant cell nests from the host tissue [13]. Shi et al. in their study reported an uneven distribution and prognostic significance of margin-infiltrating lymphocytes on both sides of the tumor capsule [14]. This observation corresponded to the results of our pilot study, which showed the presence of a discernible capsule around tumor nests in the majority of patients and unbalanced distribution of immune cells through the margin [15]. Furthermore, contradictory data have often been reported regarding the prognostic value of a particular cell type in a single region of interest (ROI) or their combinations [12,16,17]. Although T-cells were recognized as the most important players within the adaptive immunity arm, several groups have reported inconsistent data on the prognostic impact of B-cells infiltrating the tumor and the invasive margin of HCC [12,14].

We therefore hypothesized that the heterogeneity of immune cell densities through the tumor invasive margin of HCC may confer different prognostic significance, and that the assessment of B-cells, besides T-cells, may improve prognostication of unfavorable outcomes after curative HCC resection. To test this hypothesis, we evaluated the significance of CD3+ and CD8+ T-cell and CD20+ B-cell infiltration in different regions of the tumor and non-tumor liver, with the emphasize on the tumor invasive margin, as individual or associated prognostic factors for time to recurrence (TTR), DFS and OS.

## 2. Results

### 2.1. Demographics of HCC Patients

The demographics and clinical characteristics of the patients are shown in Appendix A. The median age of the patients was 69 years, and males accounted for 77.6%. Regarding the etiology of HCC, the most frequent background disease was chronic non-viral hepatitis with prevalence of non-alcoholic steatohepatitis (NASH) (23.9%). Forty-six patients, thirteen patients, five patients and three patients were at TNM stages I, II, III and IV, respectively. Pathology characteristics of the resected tumors are presented in Appendix A. Using the Edmondson–Steiner system, most of the tumors (70.1%) were histologically graded as G2. Predominant growth types were mixed (50.7%) and desmoplastic (46.3%). No evidence of extracellular matrix deposition at the tumor interface was observed in 1.5% cases, incomplete encapsulation of <50% and >50% interface was found in 13.4% and 38.8% cases, and complete encapsulation of interface in 46.3% of cases.

### 2.2. Outcomes

At the time of the last follow-up, 29 (41.8%) patients had tumor recurrence, and 38 (56.7%) patients had died. Using the Kaplan–Meier methodology, the recurrence-free proportion was 48.7%, and the estimated probability of DFS and OS were found to be 33.2% and 49.4% at 5 years, respectively (Appendix A).

### 2.3. Distribution of Immune Cells in Different Regions of Interest (ROIs)

Both CD3+ and CD8+ T-cells populations displayed significantly greater density of nucleated cell profiles (Q_A_) than CD20+ B-cells (*p* < 0.001) in each ROI (shown in Figure 1A). There was a gradient of Q_A_ of CD20+ B-cells, CD3+ and CD8+ T-cells between ROIs from the highest in peritumor (PT) liver through the margin to the lowest in TC (as it is demonstrated for CD20+ cells on Figure 2A). The outer margin displayed remarkably greater Q_A_ of all cell types in the vast majority of patients compared to the inner margin (shown in Figure 1B, *p* < 0.001). We thus decided to treat the inner and outer layers of the margin as distinct ROIs. Q_A_ of CD20+ (*p* < 0.01), CD3+ (*p* < 0.001) and CD8+ cells (*p* < 0.001) were smaller in TC compared with the inner margin; however, they did not differ significantly between the outer margin and PT liver (*p* > 0.05). In non-tumor (NT) liver, Q_A_ of CD20+ cells were greater compared with TC (*p* < 0.01), however Q_A_ of all cell types were smaller than in the margin and PT liver (*p* < 0.05) (shown in Figure 1A).

Q_A_ of CD3+, CD8+ and CD20+ cells were more than ten times smaller in the inner margin compared with the outer margin in 9%, 13%, and 46% of patients, respectively (shown in Appendix A).

### 2.4. Prognostic Values of Immune Cells

We analysed the associations between the Q_A_ of immune cells in each ROI and TTR, DFS and OS. None of the clinical, pathology or immunohistochemistry (IHC) variables correlated with OS. Among the clinical and pathology variables, younger age and higher TNM tumor stage were associated with greater risk of recurrence, whereas no variable was associated with DFS (Appendix A).

Both intermediate and high Q_A_ of CD8+ cells in TC and of CD20+ cells in the inner margin (as it is depicted for CD20+ cells in Figure 3) were associated with low risk of recurrence (Table 1). Only high Q_A_ of CD3+ and CD20+ cells in TC, of CD3+ and CD8+ in the inner margin, as well as intermediate Q_A_ of CD20+ cells in PT liver, were associated with low risk of recurrence. These findings were confirmed in the Kaplan–Meier analysis (shown in Appendix A), which also showed a longer TTR in patients with high vs. intermediate Q_A_ of CD3+ and CD20+ cells in TC (*p* = 0.047 and *p* = 0.023, respectively) and inner margin (*p* = 0.005 and *p* = 0.017, respectively).

Both intermediate and high Q_A_ of CD20+ cells in the inner margin and of CD8+ in the outer margin, high Q_A_ of CD3+, CD8+ and CD20+ cells in the TC, high Q_A_ of CD3+ and CD8+ cells in the inner margin, and intermediate Q_A_ of CD20+ cells in the PT liver were associated with longer DFS (Table 1). Kaplan–Meier analysis also demonstrated a longer DFS in patients with high vs. intermediate Q_A_ of CD3+, CD8+ and CD20+ cells in TC (*p* = 0.040, *p* = 0.043 and *p* = 0.019, respectively) and inner margin (*p* = 0.004, *p* = 0.004 and *p* = 0.014, respectively) (shown in Appendix A).

Combined intermediate-high versus low Q_A_ of CD8+ T-cells in the TC demonstrated associations with TTR (hazard ratio (HR) = 0.34, confidence interval (CI): 0.15–0.74, *p* = 0.007) and DFS (HR = 0.43, CI: 0.22–0.83, *p* = 0.012). Intermediate-high Q_A_ of CD20+ B-cells in the inner margin were also associated with TTR (HR = 0.26, CI: 0.11–0.61, *p* = 0.002) and DFS (HR = 0.27, CI: 0.13–0.54, *p* < 0.001).

To decrease the possible effects of the long duration of the study, we excluded three patents who were operated on before 2005. This gave us a cohort size of 64 patients and reduced the duration of the study to 15 years, with a median follow-up of 73 (95%CI: 40–106) months. The associations between the Q_A_ of immune cells in each ROI and TTR in terms of HRs and *p*-values (provided in Appendix A) were very close to that obtained for the whole cohort for Q_A_ per individual region of interest (Table 1). As for clinical and pathology variables, association of age with TTR remained significant in the cohort of 64 patients, whereas the effect of the TNM stage was borderline (Appendix A).

Intermediate and high Q_A_ of CD8+ cells in the entire margin were associated with lower risk or recurrence and longer DFS (Appendix A). No cell type in the NT liver showed associations with TTR or DFS (Appendix A).

We performed univariable Cox regression using ratios of Q_A_ of immune cells between different ROIs that were dichotomized at the median level (Table 2). Above-median Q_A_ ratio of CD20+ cells between inner and outer margins had the lowest HRs and *p*-values in predicting TTR and DFS. Above-median inner/outer margin Q_A_ ratio of CD3+ cells was associated with both longer TTR and DFS, whereas above-median Q_A_ ratio of CD8+ cells predicted only longer DFS. Our findings were confirmed in Kaplan–Meier analysis (shown in Appendix A).

Taking into account the highlighted importance of the inner margin, we then explored whether the combination of CD8+ and CD20+ cells can improve the prognostic significance of each other taken alone. Kaplan–Meier analysis (shown in Figure 4) demonstrated longer TTR and DFS in patients with intermediate-high Q_A_ of both markers in the inner margin compared with only CD20+ intermediate-high (*p* = 0.001 and *p* = 0.027 for TTR and DFS, respectively), only CD8+ intermediate-high (*p* < 0.001 and *p* < 0.001) and CD20+ low/CD8+ low (*p* = 0.030 and 0.006), whereas in the TC combination CD8+ and CD20+ cells did not improve prognostic value (data not shown).

Multivariable Cox regression models for each immunohistochemical parameter adjusted for TNM staging and age were built to assess their independent ability to predict TTR (Table 3). Among the Q_A_ of individual cells, both intermediate and high CD8+ in TC were independently associated with TTR. Only high Q_A_ of CD3+ and CD20+ in TC and high Q_A_ of all cell types in the inner margin, as well as intermediate Q_A_ of CD20+ cells in PT liver, were also independently predictive of low risk of recurrence. Ratios of CD3+ and CD20+ Q_A_ between inner and outer margin and ratios of CD20+ Q_A_ between TC and outer margin or PT liver all retained significant associations with TTR.

### 2.5. Association between Subtypes of Immune Cells and Clinical and Pathology Variables

For all cell types, the strongest correlations were observed between their Q_A_ in TC and inner margin (Appendix A). For each cell type, the inner/outer margin Q_A_ ratio correlated with respective cell Q_A_ in the TC: ρ = 0.50 for CD3+, ρ = 0.63 for CD8+ and ρ = 0.47 for CD20+ cells (*p* < 0.001).

Q_A_ of CD3+, CD8+ and CD20+ cells correlated between each other within each ROI (Appendix A).

Appendix A shows correlations of Q_A_ of CD3+, CD8+ and CD20+ cells with clinical and pathology variables. High Q_A_ of CD3+, CD8+ and CD20+ cells in TC and high Q_A_ of CD3+ and CD8+ cells in the inner margin were observed in less differentiated tumors since they correlated with greater Edmondson–Steiner grading score and nuclear grades. High amounts of stroma inside tumor nests correlated with high Q_A_ of CD3+, CD8+ and CD20+ cells in TC and high Q_A_ of CD3+ and CD8+ cells in the inner margin, indicating stromal localization of TIL. Inner/outer margin Q_A_ ratios of T- and B-cells did not correlate with any clinical or pathology variables, including the extent of tumor encapsulation (data not shown).

Q_A_ of immune cells in the PT liver did not correlate with any of the tumor pathology features, however high Q_A_ of CD3+ cells correlated with higher grade of chronic hepatitis and high Q_A_ of CD8+ cells correlated with higher grades of NASH and chronic hepatitis (Appendix A). High Q_A_ of CD3+ and CD20+ cells in the NT liver correlated with higher grades and stages of NASH and chronic hepatitis.

## 3. Discussion

In our paper, we analysed the immune microenvironment of HCC after a curative resection in terms of type, density, location and eventual interaction of adaptive immune cells within distinct tumor regions. For the first time, we examined the tumor margin as two separate entities: inner and outer margin, which demonstrated statistical differences in immune cell infiltration with different prognostic significance. The inner margin harbored smaller Q_A_ of CD3+ and CD8+ T-cells and CD20+ B-cells than the outer margin. However, in contrast to the outer margin, high Q_A_ of T-cells and intermediate and high Q_A_ of B-cells in the inner margin strongly predicted longer TTR and DFS. High inner/outer margin Q_A_ ratios of T- and B-cells were also associated with longer TTR and DFS. By contrast, low cells’ Q_A_ and ratios were associated with worse TTR and DFS. Patients with intermediate-high Q_A_ of combined CD8+ and CD20+ cells in the inner margin showed superior TTR and DFS, compared to CD8+-high or CD20+-high alone. In TC, intermediate and high Q_A_ of CD8+ cells predicted longer TTR, whereas CD3+ and CD20+ were predictive only at high Q_A_. DFS was predicted by high Q_A_ of CD3+, CD8+ and CD20+ cells in TC. In view of the lower importance of CD3+ T-cells, we will further discuss only CD8+ and CD20+ cells.

### 3.1. Heterogeneity of the Tumor Invasive Margin

We believe that greater infiltration of the outer 500 μm of invasive margin compared with the inner 500 μm might be partially attributed to the encapsulation of the tumor nests present in a vast majority of the patients in our cohort (Appendix A). The observed frequency of encapsulation is in agreement with the literature [18,19]. The histological pattern of the tumor margin corresponded to its macroscopic description: 62.7% of patients presented with a single distinctly nodular tumor, which is usually characterized by the presence of a discernible capsule [18]. The encapsulation in primary and metastatic HCC has been reported to be associated with decreased invasiveness and improved survival [19,20,21]. However, contrary to immune cells’ parameters, encapsulation was not associated with outcomes in our study (Appendix A). Inner/outer margin ratios of Q_A_ of immune cells were also not associated with tumor encapsulation, therefore, the fibrotic capsule itself is not a serious obstacle for cell penetration into the tumor. Instead, the characteristics of the tumor and immune cells, as well as other players in the tumor microenvironment, might be more important mechanisms regulating cell traffic, however, all those hypotheses are awaiting testing in HCC [22,23].

### 3.2. CD8+ Cells in TC and CD20+ Cells in the Inner Margin Confer the Best Prediction of TTR and DFS

High immune infiltration of TC and the inner margin correlated with tumor de-differentiation (Appendix A), which was shown previously in breast cancer [24] but not in HCC [25]. This observation implies substantial antigenicity of HCC for the initiation of immune response. Our data also propose a major importance of CD8+ in TC and CD20+ cells in the inner margin for generating memory response, as they conferred lower risk of recurrence and longer DFS at both intermediate and high Q_A_ in those ROIs, whereas low cell Q_A_ conferred shorter TTR and DFS, accordingly. Patterns of TIME with high vs. low infiltration of the interior of the tumor are referred to in the literature as “hot” or “immune-inflamed” and “cold” or “immune-desert”, and are associated with a better and worse prognosis, respectively [22,26].

### 3.3. CD8+ and CD20+ Cells in the Inner Margin Cooperate in Initiating Antitumor Immune Response

The individual predictive abilities of CD8+ T-cells and CD20+ B-cells in the inner margin increased when we combined them. Patients with intermediate-high Q_A_ of both markers in the inner margin (but not in the TC) had the best TTR and DFS, which were significantly longer than in only CD20+high and only CD8+high groups, whereas patients with low Q_A_ of both markers displayed the worst TTR and DFS. Similar findings were reported earlier by Shi et al. [14], who demonstrated longer TTR and OS in patients with both high CD8+ and CD20+ cell densities in the entire invasive margin of HCC. Mlecnik et al. [27] concluded that the TB-score, which is based on the assessment of CD8+ T-cells and CD20+ B-cells, along with Immunoscore in liver metastases of colorectal cancer were the strongest predictors of DFS and OS. Tumor-infiltrating CD8+ T-cells are thought to represent the effector memory phenotype, which is considered the main anti-tumor actor in HCC and many other cancers [9,26]. Tumor-infiltrating B-cells can exert their prolonged anti-tumor impact as antigen-presenting cells that trigger and modulate T-cell immune response, through production of antibodies against tumor antigens, and by the direct killing of cancer cells (reviewed in [28]). Along with the positive correlation between CD8+ and CD20+-cells in all ROIs, our results suggest that these cells cooperate in the mounting of a long-lasting memory response, able to limit local or systemic tumor dissemination [10,29].

### 3.4. Greater T- and B-Cell Ratios between Inner and Outer Margin Associate with Longer TTR and DFS

In addition to individual and combined cell Q_A_ in the TC and inner margin, above-median inner/outer margin Q_A_ ratios were strongly associated with longer DFS and TTR for CD20+ cells, and longer DFS for CD8+ T cells. On the contrary, under-median ratios conferred a worse prognosis. Inner/outer margin ratios correlated with respective cell Q_A_ in the TC, therefore, patients with above-median ratios apparently had “hot” tumors and vice versa. We propose that the Q_A_ ratios between the inner and outer margin, which may be a measure of cell penetration through the tumor border, could help to better characterize TIME phenotypes and risk stratification. “Poor” (<0.1) inner/outer margin Q_A_ ratio corresponds to the immune-excluded phenotype, which reflects the effective mounting of the host T- and B-cell immune response, along with the ability of the tumour to escape such a response [22]. To our knowledge, only one previous HCC study [14] has explored the impact of penetration of CD20+ lymphocytes into the tumor, which was accompanied by better survival. In our paper, we expanded this finding to T-cells and quantified the extent of cell penetration into the tumor. Despite their very limited ability to pass through the tumor border, CD20+ cells in the inner margin predicted TTR and DFS at the lowest HRs and *p*-values and conferred better survival, not only at high but at intermediate and intermediate-high Q_A_ as well, which points at the importance of B-cells. Of note, the immune-excluded and cold phenotypes are accompanied by resistance to therapy with immune-check-point inhibitors [22,26]. For a mechanistic basis, it has been shown that blockade of TGFβ signalling or vascular-targeted therapy in preclinical models permitted lymphocyte infiltration into the tumor [23,30].

### 3.5. Comparison to Relevant Literature

About 60 papers assessing the prognostic value of TIL in HCC had been published as of 2018 (reviewed in [12]), however, a vast majority of these assessed cell infiltration in TC. In general, the pooled results correspond to our findings, showing better DFS in HCC patients with greater CD8+ cell densities in TC. Only a minority of previous studies aimed at evaluating TIL in the entire margin of HCC and demonstrated the positive impact of CD3+ [17] and CD8+ cells on DFS [17,31], and CD20+ on TTR and OS [14]. Several groups reported results discordant with ours. Guiscia et al. [32] did not observe significant associations of intratumoral CD8+ T-cell with progression-free survival in HCC. Zheng and co-authors [33] found an association of high density of CD3+ or CD8+ T-cells in the TC, but not in the margin, and of CD8+ cells in PT liver with better OS after resection of HCC. Ramzan et al. [34] correlated high densities of CD8+ T-cells in cirrhotic parenchyma, but not in TC with recurrence and lower OS after resection. Sun et al. [17] stated that T-cells in TC were more positively influential on DFS and OS than those in the margin region. These discrepancies can be related to the study of cohorts of different sizes and ethnicity, with a varying percentage of patients with cirrhosis or viral hepatitis, and several other factors, which were highlighted in the systematic review by Yao et al. [35]. Another source of the discordant results might be different or poor annotation of the margin, PT and NT liver, and assessment of cell numbers per several representative high-power FOV, instead of estimating their densities [17,32,33]. We believe that adherence to the recommendations of international consortium on assessment of TIL [13] might improve the reproducibility of the reported data.

### 3.6. Limitations

Our study has several limitations, the small sample size being the main one. Low prevalence of hepatitis and cirrhosis may not allow the reproduction of data on other cohorts, such as Asian patients. On the other hand, our study provides insight into TIME of the HCC in patients with steatohepatitis as a background disease. A relatively low rate of deaths, along with the small sample size, did not allow us to demonstrate associations of any variable with OS; however, meta-analysis by Ding et al. [12] also showed frequent absence or even inverse associations of immune cells densities with OS, compared to DFS. The death of a patient in the late postoperative period can often be caused by a number of reasons not directly related to the primary HCC itself. In our study, we aimed at assessing the role of adaptive immunity, however, it should also be noted that innate immune cells can control cancer directly by interacting with tumor cells, and/or indirectly by favoring the anti-tumor activities of CD8+ T-cells [9,23,26]. Currently, a more extensive inclusion of other players in TIME is ongoing in our laboratory.

## 4. Patients and Methods

We conducted a single-center retrospective cohort study of 70 consecutive patients with pathologically confirmed HCC stages I to IV, who underwent tumor resection at the Pilsen University Hospital (Czech Republic) between 1997 and 2019. Two of them were operated for recurrence of HCC. All pathology reports were reviewed. None of the patients had distant metastasis or received any neoadjuvant therapies such as radiotherapy and chemotherapy before operation. After exclusion of three patients with low quality of histological samples, the remaining 67 patients were included into the study (Appendix A). This retrospective research was carried out in accordance with the ethical standards laid down in the Declaration of Helsinki (2013 version); it was approved by the Ethical Board of Faculty of Medicine and University hospital in Pilsen (118/2021, 11 March 2021). Clinical tumor stage was determined according to the 8th edition of the *American Joint Commission on Cancer* [7].

Patients were followed until December 2020, with a median observation time of 101 (confidence interval: 59–143) months (assessed by the inverse Kaplan–Meier method). A diagnosis of recurrence was based on typical imaging appearance in computed tomography and/or magnetic resonance imaging scan and an elevated α-fetoprotein level. Post-operative treatment was in accordance with generally accepted guidelines.

### 4.1. Pathology and Immunohistology

For each patient, 2 to 4 blocks of formalin-fixed paraffin-embedded tissue containing the center of the tumor, invasive margin and separate block with non-tumor liver, 2- to 3 cm distant from the tumor site (if available), were retrieved from the pathology archive. Hematoxylin and eosin and Masson’s trichrome stained sections were used to evaluate the histopathology features of tumor and non-tumor tissue. The Edmondson–Steiner and WHO grading systems were used to assess tumor differentiation [36] (Appendix A). In addition, architectural, nuclear and nucleolar grades were scored [36]. Desmoplastic, pushing or infiltrative growth type [21], cytologic type, micronodularity, presence of microsatellites, microvascular invasion, and necrosis were also recorded [37]. Amount of stromal component inside the tumor was assessed on a semiquantitative scale (0–3). The extent of encapsulation was graded according to [20] as 0 (no evidence of extracellular matrix deposition at tumor interface), 1 (incomplete encapsulation, at <50% of interface), 2 (incomplete encapsulation >50% of interface, or 3 (complete encapsulation). One or two tissue sections of 4-μm thickness were cut and mounted onto BOND Plus Microscope Slides (Cat# 00270, Leica Biosystems Newcastle Ltd., Newcastle, UK). Immunohistochemical detection of CD3+ T cells, CD8+ T cells and CD20+ B cells was performed using fully automated BOND-III IHC/ISH stainer. Ready-to-use monoclonal primary antibodies for CD3 (clone LN10), CD8 (clone 4B11) and CD20 (clone L26) all from Leica Biosystems (Newcastle Ltd., UK) were used. Binding of primary antibodies with cell membranes was visualized using horseradish peroxidase (HRP)-linker antibody conjugate system (Bond™ Polymer Refine Detection). Sections were counterstained with Mayer’s hematoxylin and embedded into Micromount mounting medium (Leica Biosystems Newcastle Ltd., UK). Appropriate positive (tonsils) and negative tissue control samples were used throughout. Non-tumor liver was examined for the features of chronic hepatitis and non-alcoholic fatty liver disease (NAFLD), including NASH. Grading and staging for NAFLD and chronic hepatitis were performed according with Brunt [38] and Ishak [39] approaches, respectively.

### 4.2. Definitions of Regions of Interest

All sections were examined under Olympus CX41 microscope (Olympus, Tokyo, Japan) by two pathologists (AT and LC). To evaluate the spatial heterogeneity of immune components, the reference space within each section was microanatomically divided into TC, inner and outer invasive margin, PT liver and NT liver (shown in Figure 2 and Figure 3). The inner margin and outer margin were defined as 500 μm on each side of the border separating the malignant cell nests and adjacent non-tumor liver tissue or fibrous capsule [13] towards TC or NT liver, respectively. The TC represents the remaining tumor area. The PT region was defined as the 500 μm thick region immediately adjacent to the outer M. For each ROI (TC, inner margin, outer margin, PT liver and NT liver), eight FOV were selected by systematic uniform random sampling using the objective 20× for all ROIs except PT liver (Figure 2A). Since it was not possible to sample PT area correctly with the objective 20×, we used the objective 10×, which provided an acceptable resolution. Pictures were captured by the PromiCam 3–3CP digital camera (Promicra, Prague, Czech Republic), coupled with the QuickPhoto Industrial 3.2 software (Promicra, Prague, Czech Republic).

### 4.3. Stereological Analysis

The stereological analysis was performed by AT, EA and PK, who were blinded to the clinical outcome, using the computer assisted stereology software Ellipse (ViDiTo, Kosice, Slovak Republic). CD3-, CD8- or CD20-immunopositive nucleated cell profiles were counted using a probe consisting of a set of 2D unbiased counting frames (UCF) (Figure 2B–D). Only cells transected during cutting in the mid portion, which were found inside UCF and were not touching or being transected by exclusion lines, were counted. For TC and NT liver sizes of UCF (from 8700 to 34,801 µm^2^) and their number (6–9) per image varied according to the lymphocytic density and, therefore, sampling fraction varied from 417,611 µm^2^ to 1,670,443 µm^2^. Six UCFs per image with a total area of 417,611 µm^2^ were used for the inner and outer margin, and nine UCFs with a total area of 417,215 µm^2^ were used for the PT region. Q_A_ of CD3+, CD8+ or CD20+ cells was estimated as the number of nucleated immunopositive cell profiles divided by the total area of unbiased counting frames [40]. Inter-observer variations over 10% were re-estimated for a concord result.

### 4.4. Prognostic Factors and Outcomes

The prognostic associations of two types of immune cell parameters were explored. First, we evaluated Q_A_ of individual immune cell types in each ROI. To eliminate skewness in the distribution, the raw cell Q_A_ for the most of analyses were converted into corresponding percentile values and categorized into low (below 25th percentile) vs. intermediate (25th–70th percentile) vs. high (above 70th percentile), as well as into low vs. intermediate-high at the 25th percentile.

Second, we calculated ratios of cell Q_A_ between TC and inner margin from one side, and outer margin or PT liver from the other side that might have characterized the efficiency of penetration of immune cells into the tumor. For this purpose, we used raw cell Q_A_, and we performed dichotomization into the above-median and under-median groups. Considering the significant associations of CD20+ and CD8+ cells’ Q_A_ in the inner margin with TTR and DFS, we assessed their combined influence. To do so, patients were classified into four groups, using the 25th percentile as the cut-off: both CD20+ and CD8+ intermediate-high, only CD20+ intermediate-high, only CD8+ intermediate-high, both CD20+ and CD8+ low.

The endpoints of the study were TTR, DFS and OS. TTR was defined as the time from the date of tumor resection to the date of diagnosis of recurrence/metastasis. If recurrence was not diagnosed, patients were censored at either the date of death or the date of last follow-up. The appropriate proportion of patients without recurrence was denoted as recurrence-free proportion. DFS was considered as the time from tumor resection to the date of diagnosis of recurrence/metastasis or death due to any cause. OS was defined as the time from tumor resection to death due to any cause. Patients without recurrence or death were censored at the last follow-up.

### 4.5. Statistical Analysis

Continuous nonparametric data are expressed as median (min-max); their comparison was made either by Mann–Whitney U-test or by Friedman ANOVA, followed by Wilcoxon matched pairs test with Bonferroni correction. Proportions are expressed as raw data (percentages). The associations between pairs of ordinal or quantitative variables were assessed using Spearman correlation. To determine the prognostic value of individual predictors for TTR, DFS and OS, a univariable followed by backward stepwise multivariable Cox regression analysis was performed. Only variables that were significant in univariable analysis were included into the multivariable model [41]. Hazard ratios showing the relative risk for the intermediate or high groups separately or combined intermediate-high group or over-median group compared with 1 for the low or under-median group, were calculated.

TTR, DFS and OS were calculated by the Kaplan–Meier method and compared between groups by the log-rank test. Statistica 10 (StatSoft Inc, Tulsa, OK, USA), GraphPad Prism 9.0, (GraphPad Software LLC) and R environment (v.4.1.1) were used for statistical analyses. Kaplan–Meier analysis was performed with survival package and plots were generated with survminer package [42,43].

A 2-sided *p* value < 0.05 was considered statistically significant.

## 5. Conclusions

The overall results show distinct features of the inner and outer margin of primary HCC in most of the patients after curative resection. The inner margin, despite its smaller T- and B-cell Q_A_ compared with outer margin, conveys the best prediction for TTR and DFS. The prediction is further improved by combining CD20+ and CD8+ cells in the inner margin or by assessing inner margin/outer margin ratios. Our findings highlight first the importance of the inner margin, and secondly the predictive role of B-cells. The results should be independently confirmed, but they offer a novel approach for the stratification of patients after curative liver resection and for identifying those who are at risk for early tumor recurrence.

## Figures and Tables

**Figure 1 cancers-14-00604-f001:**
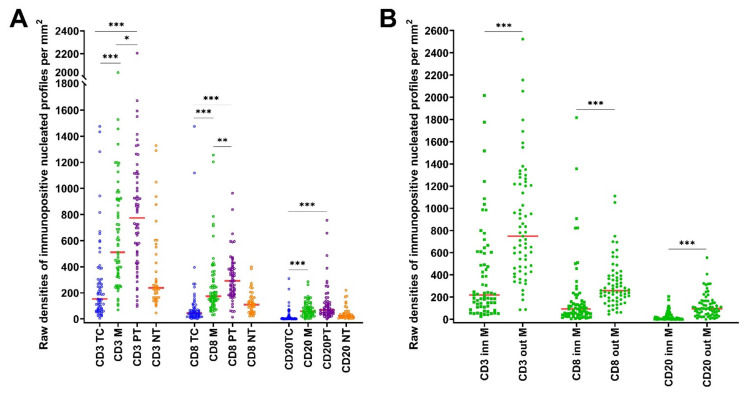
Statistics depicting the spatial distribution of nucleated profiles of CD3+, CD8+ and CD20+ tumor infiltrating lymphocytes per mm^2^ of the section (Q_A_) in the TC, M, PT liver and NT liver (**A**) and in the inner and outer tumor invasive margin (**B**). Red lines: median. *: *p* < 0.05, **: *p* < 0.01, ***: *p* < 0.001. Abbreviations: TC: tumor center, M: tumor invasive margin, inn M: inner invasive margin, out M: outer invasive margin, PT: peritumor liver, NT: non-tumor liver.

**Figure 2 cancers-14-00604-f002:**
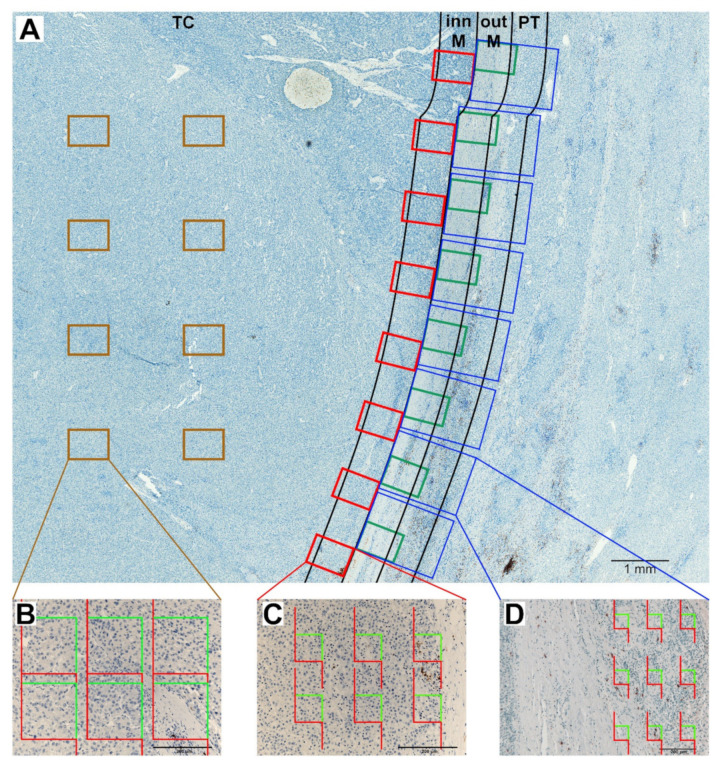
Immunoperoxidase staining for CD20+ lymphocytes in hepatocellular carcinoma. (**A**) Regions of interest (ROIs) are denoted: TC (tumor center), inn M (inner margin), out M (outer margin), PT (peritumor liver). The inn M and out M were defined as 500 µm on each side of the border separating the malignant cell nests and adjacent non-tumor tissue. The TC represented the remaining tumor area. The PT region was defined as the 500 µm thick region immediately adjacent to the out M. Eight equidistant fields of view (FOV) were taken from each ROI using systematic uniform random sampling. To sample TC, inn M and out M objective 20× was used, whereas objective 10× was used for PT region. This figure shows an example of low density of CD20+ nucleated cell profiles in the TC and inn M. (**B**–**D**) CD20+ nucleated cell profiles were counted using sets of unbiased counting frames. Examples of counting in single FOV in the TC with low density of CD20+ nucleated cell profiles (**B**), in the inn M (**C**), and in the PT liver (**D**). Scale bars 1000 µm (**A**), 200 µm (**B**–**D**).

**Figure 3 cancers-14-00604-f003:**
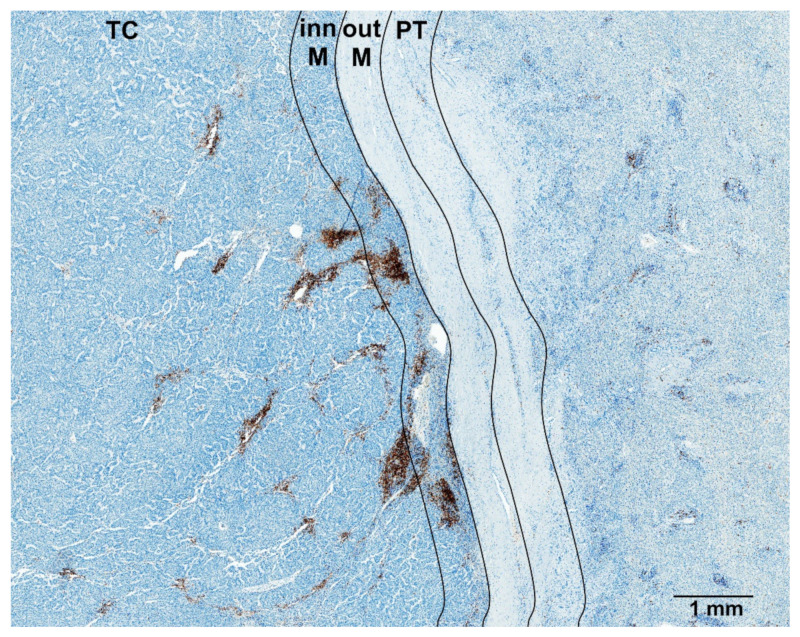
Immunoperoxidase staining for CD20+ lymphocytes in hepatocellular carcinoma that shows high density of CD20+ nucleated cell profiles (Q_A_) in the tumor center and inner margin. Regions of interest (ROIs) are denoted: TC (tumor center), inn M (inner margin), out M (outer margin), PT (peritumor liver). Scale bar 1000 µm.

**Figure 4 cancers-14-00604-f004:**
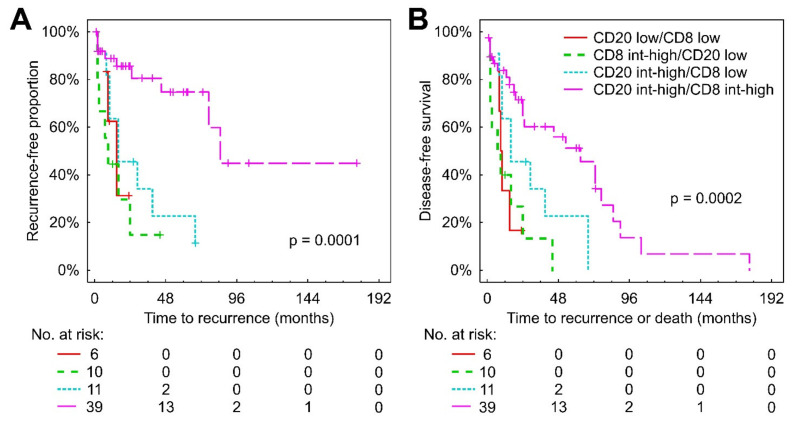
Kaplan–Meier analysis for TTR (**A**) and DFS (**B**) according to combined low vs. int-high densities of nucleated profiles of CD8+ and CD20+ lymphocytes in the inner margin. Abbreviations: int: intermediate.

**Table 1 cancers-14-00604-t001:** Densities of nucleated profiles of tumor infiltrating lymphocytes (Q_A_) per individual ROI associated with time to recurrence and disease-free survival (univariable analysis).

	Q_A_	TTR			DFS		
		HR	95% CI	*p*	HR	95% CI	*p*
**Tumor Center**							
**CD3,***n* = 67 ***p*** **= 0.030 *, 0.014** **^†^**	int vs. low	0.61	0.26–1.40	0.243	0.62	0.31–1.23	0.170
	high vs. low	0.20	0.06–0.66	**0.008**	0.28	0.12–0.66	**0.003**
**CD8,***n* = 67 ***p*** **= 0.010 *, 0.004** **^†^**	int vs. low	0.34	0.14–0.81	**0.014**	0.53	0.27–1.07	0.075
	high vs. low	0.25	0.09–0.70	**0.008**	0.23	0.09–0.55	**0.001**
**CD20,***n* = 67 ***p*** **= 0.031 *, 0.020** **^†^**	int vs. low	0.63	0.27–1.48	**0.289**	0.67	0.32–1.37	0.270
	high vs. low	0.18	0.05–0.65	**0.009**	0.27	0.11–0.69	**0.006**
**Inner invasive margin**							
**CD3,***n* = 65 ***p*** **= 0.014 *, 0.005** **^†^**	int vs. low	0.77	0.33–1.79	0.545	0.77	0.39–1.55	0.472
	high vs. low	0.14	0.04–0.54	**0.004**	0.24	0.09–0.59	**0.002**
**CD8,***n* = 66 ***p*** **= 0.029 *, 0.006** **^†^**	int vs. low	0.52	0.23–1.19	0.120	0.77	0.40–1.51	0.455
	high vs. low	0.23	0.08–0.69	**0.009**	0.22	0.09–0.57	**0.002**
**CD20,***n* = 66 ***p*** **= 0.002 *, <0.001** **^†^**	int vs. low	0.38	0.16–0.90	**0.026**	0.36	0.18–0.74	0.005
	high vs. low	0.09	0.02–0.36	**0.001**	0.13	0.05–0.35	**<0.001**
**Outer invasive margin**							
**CD3,***n* = 65 *p* = 0.354 *, 0.280 **^†^**	int vs. low	0.66	0.26–1.65	0.371	0.58	0.27–1.23	0.153
	high vs. low	0.47	0.17–1.32	0.151	0.56	0.26–1.24	0.153
**CD8,***n* = 66 *p* = 0.325 *, 0.061 **^†^**	int vs. low	0.55	0.22–1.37	0.197	0.48	0.23–0.98	**0.044**
	high vs. low	0.50	0.18–1.36	0.174	0.41	0.18–0.92	**0.030**
**CD20,***n* = 66 *p* = 0.680 *, 0.475 **^†^**	int vs. low	0.99	0.40–2.44	0.986	0.99	0.49–2.00	0.974
	high vs. low	0.68	0.25–1.87	0.452	0.65	0.29–1.45	0.290
**Peritumor liver**							
**CD3**, *n* = 64	int vs. low	0.53	0.20–1.38	0.193	0.74	0.34–1.63	0.452
*p* = 0.418 *, 0.676 ^†^	high vs. low	0.78	0.30–2.04	0.608	0.95	0.42–2.14	0.902
**CD8**, *n* = 65	int vs. low	0.82	0.33–2.04	0.663	1.09	0.52–2.26	0.828
*p* = 0.753 *, 0.952 ^†^	high vs. low	1.15	0.43–3.04	0.780	1.14	0.50–2.62	0.757
**CD20**, *n* = 65	int vs. low	0.24	0.09–0.64	**0.004**	0.43	0.20–0.92	**0.030**
***p*****= 0.014 ***, 0.053 ^†^	high vs. low	0.72	0.30–1.71	0.452	0.86	0.40–1.84	0.699

For all cells and regions of interest, the raw densities of nucleated profiles of CD3+, CD8+ and CD20+ tumor infiltrating lymphocytes per area section (mm^2^) were converted into percentiles and then categorized into low (0–25 percentile), intermediate (25–70 percentile) or high (70–100 percentile). Hazard ratios shows the relative risk compared with 1 for the low density. Bold values indicate statistical significance at the *p* < 0.05 level. * Type 3 Wald test *p* value for all 3 levels of cell densities for TTR. ^†^ Type 3 Wald test *p* value for all 3 levels of cell densities for DFS. Abbreviations: HR: hazard ratio; CI: confidence interval; TTR: time to recurrence; DFS: disease-free survival; int: intermediate.

**Table 2 cancers-14-00604-t002:** Ratios between densities of nucleated profiles of tumor infiltrating lymphocytes (Q_A_) in two ROIs associated with time to recurrence and disease-free survival (univariable analysis).

	TTR			DFS		
	HR	95% CI	*p*	HR	95% CI	*p*
**CD3+ T cells**						
Inn M/out M > 0.361 ^†^, *n* = 65	0.36	0.16–0.85	**0.019**	0.45	0.24–0.86	**0.016**
TC/out M > 0.201, *n* = 65	0.41	0.18–0.92	**0.031**	0.56	0.31–1.02	0.057
TC/M > 0.330, *n* = 65	0.53	0.24–1.17	0.117	0.64	0.35- 1.16	0.138
TC/PT > 0.219, *n* = 64	0.43	0.19–1.00	**0.050**	0.61	0.33–1.13	0.116
**CD8+ T cells**						
Inn M/out M > 0.349, *n* = 66	0.53	0.24–1.17	0.119	0.45	0.24–0.85	**0.013**
TC/out M > 0.166, *n* = 66	0.59	0.27–1.29	0.187	0.63	0.34–1.15	0.131
TC/M > 0.218, *n* = 66	0.76	0.36–1.62	0.478	0.85	0.47–1.54	0.598
TC/PT > 0.161, *n* = 65	0.41	0.18–0.93	**0.032**	0.42	0.22–0.81	**0.010**
**CD20+ B cells**						
Inn M/out M > 0.113, *n* = 66	0.28	0.12–0.65	**0.003**	0.29	0.15–0.57	**<0.001**
TC/out M > 0.051, *n* = 66	0.23	0.09–0.54	**0.001**	0.40	0.21–0.75	**0.004**
TC/M > 0.078, *n* = 66	0.24	0.10–0.57	**0.001**	0.41	0.22–0.76	**0.005**
TC/PT > 0.049, *n* = 65	0.27	0.12–0.65	**0.003**	0.49	0.27–0.90	**0.022**

For all cells and regions of interest the raw densities of nucleated profiles of CD3+, CD8+ and CD20+ tumor infiltrating lymphocytes per area section (mm^2^) were dichotomized at the median value into lower-the-median and above-the-median values. Hazard ratios shows the relative risk compared with 1 for the lower-the-median group. Bold values indicate statistical significance at the *p* < 0.05 level. ^†^ Median. Abbreviations: HR: hazard ratio; CI: confidence interval; TTR: time to recurrence; DFS: disease-free survival; TC: tumor center; M: margin; inn M: inner invasive margin; out M: outer invasive margin; PT: peritumor liver; NT: non-tumor liver.

**Table 3 cancers-14-00604-t003:** Multivariable Cox-regression for time to recurrence adjusted for age and TNM stage.

		TTR		
		HR	95% CI	*p*
**Density of nucleated profiles of TIL (Q_A_) per ROI**				
**CD3 TC ^†^** *p* = 0.082 *	int vs. low	0.78	0.32–1.96	0.601
	high vs. low	0.26	0.07–0.89	**0.032**
**CD8 TC** ***p* = 0.010 ***	int vs. low	0.34	0.14–0.81	**0.014**
	high vs. low	0.25	0.09–0.70	**0.008**
**CD20 TC ^†^** ***p* = 0.017 ***	int vs. low	0.91	0.35–2.35	0.842
	high vs. low	0.18	0.05–0.64	**0.008**
**CD3 inn M ^†^** ***p* = 0.014 ***	int vs. low	1.02	0.41–2.53	0.961
	high vs. low	0.16	0.04–0.62	**0.008**
**CD8 inn M ^†^** *p* = 0.065 *	int vs. low	0.60	0.25–1.39	0.230
	high vs. low	0.26	0.09–0.81	**0.020**
**CD20 inn M ^†^** ***p* = 0.004 ***	int vs. low	0.50	0.20–1.27	0.147
	high vs. low	0.10	0.03–0.39	**0.001**
**CD20 PT ^†^** ***p* = 0.018 ***	int vs. low	0.24	0.09–0.66	**0.005**
	high vs. low	0.70	0.29–1.67	0.420
**Ratios between densities of nucleated profiles of TIL (Q_A_) in two ROIs**				
**CD3 inn M/out M ^ǂ^**	above vs. under median (0.361)	0.35	0.15–0.81	**0.015**
**CD3 TC/out M ^ǂ^**	above vs. under median (0.201)	0.45	0.21–1.03	0.058
**CD3 TC/PT ^ǂ^**	above vs. under median (0.219)	0.48	0.21–1.13	0.093
**CD8 TC/PT ^ǂ^**	above vs. under median (0.161)	0.48	0.20–1.12	0.091
**CD20 inn M/out M ^ǂ^**	above vs. under median (0.113)	0.26	0.11–0.62	**0.002**
**CD20 TC/out M ^ǂ^**	above vs. under median (0.051)	0.21	0.09–0.51	**0.001**
**CD20 TC/PT ^ǂ^**	above vs. under median (0.049)	0.28	0.12–0.66	**0.004**

^†^ For all cells and regions of interest, raw densities of nucleated profiles of CD3+, CD8+ and CD20+ tumor infiltrating lymphocytes per mm^2^ (Q_A_) were converted into percentiles and then categorized into low (0–25 percentile), intermediate (25–70 percentile) and high (70–100 percentile). ^ǂ^ for all cells and regions of interest raw densities of nucleated profiles of CD3+, CD8+ and CD20+ tumor infiltrating lymphocytes per mm^2^ were dichotomized at the median level. TC: center of the tumor, M: tumor invasive margin, PT: peritumor liver, inn M: inner invasive margin, out M: outer invasive margin. Bold values indicate statistical significance at the *p* < 0.05 level. * Type 3 Wald test *p* value for all 3 levels of cell densities. Abbreviations: HR: hazard ratio; CI: confidence interval; ROI: region of interest; TTR: time to recurrence; TC: tumor center; inn M: inner invasive margin; out M: outer invasive margin; PT: peritumor liver; NT: non-tumor liver; TIL: tumor infiltrating lymphocytes.

## Data Availability

All data generated or analyzed during this study are included in this article and its Appendix A. Further enquiries can be directed to the corresponding author.

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
