# Peer review of "T- and B-Cells in the Inner Invasive Margin of Hepatocellular Carcinoma after Resection Associate with Favorable Prognosis"

_cancers, 2022, doi:10.3390/cancers14030604_

Round 1
Reviewer 1 Report
The authors studied 67 patients with HCC after resection and evaluated the significance of CD3+ and CD8+ T-lymphocytes and CD20+ B-lymphocytes in tumor and non-tumor liver for TTR, DFS and OS. They showed that T- and B-cell numbers in the inner margin compared with those in the outer margin conveys the best prediction for TTR and DFS, and that the prediction is further improved by combining CD20+ and CD8+ cells in the inner margin or by assessing inner margin/outer margin ratios.
The manuscript is interesting. I have some considerable concerns.
Major comments
#1. The duration of study is 23 years. This means that they collected 2.9 patients per year which seems to be too small. The procedures of surgery, diagnostic methods, and therapies other than surgery must have changed significantly. These factors may affect the results of this study.
#2. The follow-up periods should be varied significantly. They should present the follow-up period.
#3. Table S1. Child-Pugh score is not available in 53.7% of the patients studied. I wonder how the authors evaluated the liver reserve before surgery.
#4. Tumor size is unknown in 3 patients. Can the authors collect the accurate data in these patients?
#5. In 3 patients, TNM is stage IV. Was the curative resection possible in them?
Minor comments
#1. Line 133. IHC should be spelled out, since this is the 1st appearance.
#2. The authors should use single abbreviations after 2nd appearance. For examples, they use IM, inner M, and inn M for inner margin, and outer M, out M, and OM for outer margin.
#3. Figure 1. Some bars indicating the data compared are missing.
#4. Table 1. “int” should be spelled out.
#5. Figure 4. “int” should be spelled out.
#6. Line 312. “TB-score” need explanation.
Author Response
Dear Reviewer,
Thank you very much for your comments. Please, find below our responses.
Major comments
#1. The duration of study is 23 years. This means that they collected 2.9 patients per year which seems to be too small. The procedures of surgery, diagnostic methods, and therapies other than surgery must have changed significantly. These factors may affect the results of this study.
Answer: Indeed, the first patient from the cohort was operated in 1997. However, the types of surgical procedures used throughout the study were the same.
21 patients from the cohort received palliative or adjuvant chemotherapy, however, when tested in in univariate Cox regression they showed no effect. Targeted therapy was used in 10 of them but this showed no effect in univariate Cox regression (sorafenib was given first in 2008 for patient operated in 2005).
To address your question we performed additional analyses after the exclusion of 3 patents, who were operated before 2005. It gave us the cohort size of 64 patients and reduced duration of the study to 15 years with the median follow-up 73 (95%CI: 40-106) months. The obtained results of univariable Cox regression for time to recurrence (Table S5) are very close to original data in terms of HRs and P-values for QA per individual region of interest (so data provided in the Table 1 are valid).
As for clinical and pathology variables, association of age with TTR remained significant in the cohort of 64 patients, whereas association of TNM stage with TTR was borderline (Table S6).
That is why we believe that we may keep original cohort size of 67 patients.
#2. The follow-up periods should be varied significantly. They should present the follow-up period.
Answer: We added the confidence interval to the median observation time (line 421).
#3. Table S1. Child-Pugh score is not available in 53.7% of the patients studied. I wonder how the authors evaluated the liver reserve before surgery.
Answer: Thank you for this important comment. Of course, all components of this score were evaluated routinely as a part of integral assessment in all patients before operation to assess the severity of liver disease and prognostication. However, formal records of the total score were available in digital tables in 53,7% of patients only. Now this score is provided for 97% of patients.
#4. Tumor size is unknown in 3 patients. Can the authors collect the accurate data in these patients?
Answer: data were completed for 2 of these patients (Table S1). After this tumor size remained not to be associated with any of outcomes in the study. Unfortunately, it was not possible to obtained data for 1 patient.
#5. In 3 patients, TNM is stage IV. Was the curative resection possible in them?
Answer: There are really 3 patients with T4 tumors. In one of these patients the tumor involved visceral peritoneum (Patient # 65). In two patients the tumor involved hepatic veins (Patient # 17 and Patient # 56). Whole tumor was resected in all these patients so the operation was really considered radical in all of them. Please, find below the follow-up data:
Patient # 65: resection of segments 2 and 3 in 2019, survival 10 months, still alive.
Patient # 17: extended right hepatectomy in 2009, 1 local relapse in 15 months, survival 140 months, still alive.
Patient # 56: non-anatomical resection, resection of diaphragma in 2018, survival 23 months, still alive.
Minor comments
#1. Line 133. IHC should be spelled out, since this is the 1st appearance.
Answer: corrected
#2. The authors should use single abbreviations after 2nd appearance. For examples, they use IM, inner M, and inn M for inner margin, and outer M, out M, and OM for outer margin.
Answer: corrected
#3. Figure 1. Some bars indicating the data compared are missing.
Answer: corrected (the problem appeared due to compression of the original image in the Microsoft Word).
#4. Table 1. “int” should be spelled out.
Answer: shortening was added to the footnotes
#5. Figure 4. “int” should be spelled out.
Answer: shortening was added to the footnotes
#6. Line 312. “TB-score” need explanation.
Answer: corrected
Reviewer 2 Report
The manuscript entitled “T- and B-cells in the inner invasive margin of hepatocellular 2 carcinoma after resection associate with favorable prognosis” by Andriy Trailin, et al reporting the differential distribution of T- and B-cell QA in the inner margin and the outer margin of HCC tumor resections. This study highlights the predictive and prognostic significance of spatial distribution of CD20+ and CD8+ cells either alone in combination in HCC tumor resections. These immune signatures could be a “potential biomarker” in identifying patients who might be at greater risk for recurrence. These insights may help identify patients after curative liver resection who likely would benefit from timely interventions with treatment modalities. This study was well conducted in a cohort of 67 patients, results offer a novel approach in the stratification of patients for therapeutic interventions and manuscript was well written.
But there were grammatical mistakes and spell check. Otherwise manuscript is suitable for publication. My only concern was that at the time of last follow up 56.7% of the patients (39) died, which may be pointed out as one of the limitations of the study.
Author Response
Dear Reviewer,
Thank you very much for your comments. Please, find below our responses.
Manuscript was read by native speaker and some mistakes were corrected.
The 1-, 3-, and 5-year OS rates in our study were 88.1%, 70.6 % and 49.4%, respectively, which, along with small sample size, might have not allowed us to find any predictive variable for OS. According to your suggestion we cited this as an additional limitation of the study. However, observed rates are similar or even lower that reported in the literature: the 1-, 3-, and 5-year OS rates were 84%, 64%, and 50% (Zhu et al, 2008), 83%, 59%, and 45% (Jia et al, 2010), 89%, 66%, and 44% (Shi et al, 2013), 78.2%, 57.5%, and 45.9% (Zhou et al, 2019), 82% and 57% at 1 and 3 years (Gabrielson et al, 2016).
Reviewer 3 Report
In their paper entitled "T- and B-cells in the inner invasive margin of hepatocellular carcinoma after resection associate with favorable prognosis" Trailin and coll. provide a retrospective evaluation of pathological characteristics of resected HCC, revealing an association between the type and patterno of lymphocyte distribution inside the tumor and the oncological outcomes of HCC
The implication of immune system on the oncological outcome of HCC is an intriguing field of research, that has recently gained much popularity thanks to the deveolpment of ICI for targeted therapies
The manuscript is generally well written and organized
I have few suggestions in order to improve the paper quality:
1) better discuss the impact of NLR onpostoperative outcomes of LR or LT for HCC: 10.2147/JHC.S86792
2) provide MELD score and aFP of the patient cohort
3) provide a multivariable cox regression analysis for DFS and OS including other risk factors such as MVI, aFP (important missing), satellitosis, possibly referring to rhis recent publication: 10.3390/diagnostics12010160
Best regards
Author Response
Dear reviewer,
Thank you very much for your comments. Please, find below our responses.
1) better discuss the impact of NLR on postoperative outcomes of LR or LT for HCC: 10.2147/JHC.S86792
1) NLR is a non-invasive surrogate test, which demonstrated stronger associations with outcomes after liver transplantation than after resection. In our study we aimed at assessing tumor immune microenvironment in the resected tumor and non-tumor tissue. That is why we have inserted a statement of NLR to the Introduction section only. Thank you for the reference, which we cited (reference 6).
2) provide MELD score and aFP of the patient cohort
2) Since the primary use of the MELD score is in prioritizing patients on the waiting list for liver transplantation, which we do not perform in our center, the MELD score was not calculated on the regular basis, and it is missing in 34 patients. AFP was also not measured as a standard parameter, and data are missing in 22 patients. Taking into account your suggestions, we added AFP data into Tables S1, S4 and S6.
3) provide a multivariable cox regression analysis for DFS and OS including other risk factors such as MVI, aFP (important missing), satellitosis, possibly referring to this recent publication: 10.3390/diagnostics12010160
3) Thank you for the reference, which we cited (reference 5). However according to the commonly applied statistical guidelines (see for example K.J. Rothman and S.Greenland, Modern epidemiology, Lippincott-Raven, 1998) only variables which are significant in univariate analysis should be included in multivariable models. This is particularly relevant to small sample sets like ours. We note that the Italian study that was referred to above essentially also followed these guidelines even though the sample size was 3 times higher than ours. In our univariate analysis microvascular invasion, presence of microsatellits and alfa-fetoprotein levels were not individually predictive for any of outcomes (see Table S4). They were thus not entered into multivariate analysis.
Round 2
Reviewer 1 Report
In the revised version of their manuscript, the authors have properly answered to the points raised by the reviewer.
Reviewer 3 Report
The Authors properly replied to previous comments
The paper is suitable for publication
Congratulations, and best regards